# Air Pollution and Atopic Dermatitis, from Molecular Mechanisms to Population-Level Evidence: A Review

**DOI:** 10.3390/ijerph20032526

**Published:** 2023-01-31

**Authors:** Raj P. Fadadu, Katrina Abuabara, John R. Balmes, Jon M. Hanifin, Maria L. Wei

**Affiliations:** 1Department of Dermatology, University of California, San Francisco, CA 94115, USA; 2Dermatology Service, San Francisco VA Health Care System, San Francisco, CA 94121, USA; 3School of Public Health, University of California, Berkeley, CA 94720, USA; 4Division of Occupational and Environmental Medicine, University of California, San Francisco, CA 94143, USA; 5Department of Dermatology, Oregon Health & Science University, Portland, OR 97239, USA

**Keywords:** air pollution, pollution, air pollutants, particulate matter, atopic dermatitis, eczema, review, wildfires, environment

## Abstract

Atopic dermatitis (AD) has increased in prevalence to become the most common inflammatory skin condition globally, and geographic variation and migration studies suggest an important role for environmental triggers. Air pollution, especially due to industrialization and wildfires, may contribute to the development and exacerbation of AD. We provide a comprehensive, multidisciplinary review of existing molecular and epidemiologic studies on the associations of air pollutants and AD symptoms, prevalence, incidence, severity, and clinic visits. Cell and animal studies demonstrated that air pollutants contribute to AD symptoms and disease by activating the aryl hydrocarbon receptor pathway, promoting oxidative stress, initiating a proinflammatory response, and disrupting the skin barrier function. Epidemiologic studies overall report that air pollution is associated with AD among both children and adults, though the results are not consistent among cross-sectional studies. Studies on healthcare use for AD found positive correlations between medical visits for AD and air pollutants. As the air quality worsens in many areas globally, it is important to recognize how this can increase the risk for AD, to be aware of the increased demand for AD-related medical care, and to understand how to counsel patients regarding their skin health. Further research is needed to develop treatments that prevent or mitigate air pollution-related AD symptoms.

## 1. Introduction

Air pollution is a complex mixture that includes certain solid particles, liquid droplets, and gaseous molecules; the U.S. Environmental Protection Agency has identified criteria air pollutants that are common in the U.S. and have well-known adverse human health effects, including ozone (O_3_), particulate matter (PM), carbon monoxide (CO), sulfur dioxide (SO_2_), and nitrogen dioxide (NO_2_) [1]. Solid particles, often including ultra-fine particles smaller than 0.1 microns, are found in dust, smoke, and soot [1]. Air pollutants negatively affect the functions of multiple organs, notably the heart and lungs [2]. Concentrations of certain air pollutants are increasing in areas around the world due to environmental changes, such as increased wildfires, and sociocultural trends, such as urbanization and industrialization [2]; much of the global increase in air pollution is due to more motor vehicles and coal-fired power generation, and it is a pervasive public health issue due to its many negative effects on human health. Additionally, air pollution from a California wildfire was recently found to be associated with increased clinic visit rates for itch and atopic dermatitis (AD), a common inflammatory skin disease that affects up to 1 in 5 children and 1 in 10 adults worldwide [3,4,5].

Human skin is in constant contact with the environment, and air pollutants can directly harm skin barrier function and homeostasis to contribute to the development and exacerbation of cutaneous diseases [6,7,8]. AD, often referred to as eczema, is an inflammatory skin disease in which patients have underlying skin barrier defects and a heightened immune response to irritants and allergens [9,10]. Climatic variables—such as temperature, humidity, pollen load, and sun exposure—have been shown to affect AD symptoms [11,12,13].

The effects of air pollution on AD have been less studied compared to their impact on respiratory and cardiovascular diseases, but some of the underlying pathologic mechanisms, such as triggering inflammatory responses, are similar [14,15,16]. There have been many reviews on the topic of air pollution and skin health, but most were not specific to AD, did not include recent studies, or focused on either only biological or only epidemiologic evidence [6,7,8,15,17,18,19,20,21,22]. Here, we provide an updated, comprehensive, and synthesized overview of the molecular, cellular, and epidemiologic literature focusing on the relationship between air pollution and AD. The aims of this manuscript are to discuss the connection between air pollution and AD through (i) reviewing potential biological mechanisms, (ii) evaluating population-level evidence, including impacts on healthcare utilization that were not included in previous reviews, and (iii) discussing avenues for future research.

## 2. Review of the Evidence

Many studies have been conducted around the world using a variety of exposure and outcome assessment methodologies to investigate the effects of air pollution on AD. In aggregate, they suggest a higher risk for AD symptoms and disease incidence associated with air pollution exposure. Characteristics of air pollution exposure, including the type of air pollutant, source of air pollution, and concentration of exposure (all of which can affect its impact on the skin), have been found to vary across studies.

### 2.1. Molecular Pathogenesis

Multiple biological mechanisms by which air pollution may induce or exacerbate AD have been identified, including activation of the aryl hydrocarbon receptor pathway, promotion of oxidative stress, impairment of the skin barrier, and initiation of a proinflammatory response.

#### 2.1.1. Aryl Hydrocarbon Receptor Pathway

Polycyclic aromatic hydrocarbons (PAHs) are produced from the combustion of any carbon-based fuel and are potent inflammation-inducing ligands of the aryl hydrocarbon receptor (AhR) [23]. AhR is a cytosolic ligand-activated transcription factor that can activate the expression of genes related to cell proliferation, detoxification, inflammation, and melanogenesis [23]. Transgenic mice that constitutively expressed AhR in keratinocytes had phenotypic characteristics similar to that of AD: pruritus, skin inflammation, and skin barrier dysfunction [9,24,25]. PAHs activated the AhR pathway to enhance artemin expression and inflammatory processes in murine epidermal cells [24]. Artemin is a neurotrophic factor that causes a hypersensitivity to itching, is coded by Artn—an AhR target gene—and expressed at high levels in patients with AD [26]. Human cultured epidermal cells exposed to diesel exhaust particles upregulated ARTN mRNA through activation of the AhR pathway [24]. In addition, cultured human keratinocytes exposed to O_3_ and PAHs have an increased expression of cytochrome P450 isoforms—CYP1A1, CYP1A2, and CYP1B1—through an AhR-dependent mechanism [27], indicating that the AhR pathway may be implicated in inducing a response to adverse effects of air pollution on skin cells. On the other hand, PAH-containing coal tar has therapeutic benefits in treating AD and psoriasis, suggesting that AhR activation can be beneficial for treating some existing cutaneous symptoms [28]. This is due to the nature of AhR signaling in skin: patients with AD have dominant non-canonical AhR signaling that is pathologic, so AhR agonists are helpful for restoring the balance between canonical and non-canonical pathways in these patients [29]. The activation of the AhR pathway may also contribute to greater aldo-keto reductase expression, which affects mast cell activation and stimulates a Th2 response [30].

#### 2.1.2. Oxidative Stress

Components of air pollution, such as nitrogen oxides, O_3_, and PAHs, can trigger the generation of reactive oxygen species (ROS) [31]. One mechanism involves intracellular metabolic pathways that convert PAHs into quinones, generating superoxide anion, hydrogen peroxide, and ROS [32]. Carbonyl moieties, a marker of direct oxidative damage of proteins, were analyzed in skin biopsies from 75 AD patients, and they were found to be significantly increased in AD lesions, especially in the superficial stratum corneum, compared to the skin of control patients [31]. The authors suggested that the skin of patients with AD is more susceptible to externally induced ROS damage, which can further disrupt the skin barrier function and exacerbate AD.

Another study assessed the impact of ambient O_3_ on an indicator of oxidative damage, malondialdehyde (MDA), on mouse skins [33]. Higher MDA concentrations extracted from O_3_-exposed skin, compared with covered skin, indicated that O_3_ can initiate lipid peroxidation reactions in cutaneous tissue and that O_3_ exerts its antioxidant-depleting effects through direct action on skin cells as opposed to the action of secondary products of O_3_ interactions formed in the lungs and then brought to the skin via blood [33]. Short-term O_3_ exposure also depleted levels of the antioxidants vitamin C, uric acid, and glutathione in the murine stratum corneum, indicating oxidative damage [34].

#### 2.1.3. Skin Barrier Function

Exposure to air pollution components can affect epidermal barrier function [35,36], as measured by transepidermal water loss (TEWL). Study subjects with AD exposed acutely to ambient NO_2_ were found to have increased TEWL, an indication of barrier impairment, compared to both preexposure AD skin and to post-exposure healthy skin [37]. The mechanisms underlying skin barrier disruption may be the pollution-related generation of free radicals and lipid peroxidation of polyunsaturated fatty acids in cell membranes, as is found in other organ systems [37,38]. The skin barrier function is impaired by particulate matter exposure due to the decreased expression of E-cadherin and structural proteins in the stratum corneum, such as cytokeratin and filaggrin [39], as well as increased production of matrix metalloproteinases [40,41].

#### 2.1.4. Inflammation

Studies conducted with keratinocytes from mice and humans illustrate how pollutants can facilitate the process of cutaneous inflammation. After exposure to diesel exhaust particles at high concentrations, mouse epidermal cells increased the expression of NF-κB [42], which can facilitate cytokine expression. Ushio et al. incubated cultured human keratinocytes with different concentrations of diesel exhaust particles and found a greater production of IL-1β at high exposure levels [43]. IL-1 is implicated in the pathogenesis of AD in patients with an associated FLG gene mutation [44]. In another study, researchers exposed cultured human epidermal cells to Asian dust storm particles that contained a mixture of particulate matter for 24 h [45]. The exposed cells significantly upregulated mRNA expression for several proinflammatory cytokines, including IL-6 [45], another cytokine implicated in the AD disease course [46,47,48]. These studies suggest that high levels of air pollution can induce a proinflammatory cytokine state in the skin due to cytokine release from keratinocytes [33,38,44,49].

In a study of children, researchers found a strong positive association between exposure to indoor volatile organic compounds (VOCs) and the percentage of IL-4-producing T cells in the blood of toddlers [50]. In another study, maternal exposure to VOCs was associated with a decreased percentage of interferon gamma (IFN-γ)-producing T cells and T-cell polarization toward the type 2 phenotype in neonates [51]. A reduced ability to secrete type 1 cytokines, such as IFN-γ, delays the maturation of type 1 sensitivity reactions and increases the risk for developing atopic diseases [50]. Taken together, the upregulation of proinflammatory cytokine expression and type 2 T-cell polarization may contribute to the development of symptomatic AD after air pollution exposure. In addition, the exposure of skin cells to airborne nanoparticles, such as carbonaceous pollutants, diesel exhaust particles, and tungsten carbide cobalt particles, can induce the production of proinflammatory cytokines and alter several cellular signaling pathways [52,53].

### 2.2. Population-Level Effects

Many epidemiologic studies have examined exposure to air pollutants and the development or exacerbation of AD. While the majority of studies have found a positive relationship (Table 1) [54,55,56,57,58,59,60,61,62,63,64,65,66,67,68,69,70,71,72,73,74,75,76,77,78,79,80], some have reported null or inconclusive findings (Table 2) [81,82,83,84,85,86,87,88,89,90]. The variance of the results across studies may be attributable to varying criteria for the severity and diagnosis of AD in different countries, age of the participants, and the approach to outcome assessment. In addition, people around the world are exposed to different components of air pollution at different magnitudes [91]. The heterogeneity in the type and concentration of air pollutant exposure, as well as varying cooccurring climatic factors, such as temperature and humidity, that may not be adjusted for in analyses can lead to inconsistent results across studies. A comparison of the study findings may also be limited by differences in the exposure assessment and characterization techniques; for example, ground-level monitor measurements may differ from satellite-based estimations [92]. The results across studies using self-reported data are mixed, but those with more direct measurements of the outcomes appear more likely to have positive associations. Most studies have been conducted with children as study participants, but almost all studies with adults, which are fewer in number, have found positive, though often slightly smaller, associations. Below, we focus on studies with eczema/AD as a primary outcome in children and adults, presenting data from cross-sectional studies and then longitudinal studies, followed by data on healthcare utilization.

#### 2.2.1. Air Pollution and Pediatric AD Prevalence, Incidence, and Severity

One cross-sectional study in Taiwan used a modified protocol from the International Study of Asthma and Allergies in Children (ISAAC) to examine atopic eczema symptoms in 23,980 school children using questionnaires and found that the risk of disease occurrence was strongly associated with perceived exposure to ambient air pollution, subjectively defined as not present, mild, or moderate to severe [61]. The same group later conducted a larger cross-sectional study with children and found that exposure to traffic-related air pollutants (TRAP), including CO and nitrogen oxides, was positively associated with the prevalence of flexural eczema in both boys and girls [62]. Larger associations between living near a traffic-heavy area and AD prevalence were detected in two other cross-sectional studies conducted in Bolivia [69] (odds ratio (OR) 1.4, 95% confidence interval (CI): 1.1–2.0) and Lebanon [68] (OR = 1.5, 95% CI: 1.1–2.0), though air pollution exposure was not quantified. These results are supported by studies in Korea that found significant but weaker associations for TRAP exposure among children [73,75]. Other cross-sectional analyses have found non-significant associations between various air pollutants and AD in children, many of which have also used ISAAC data and similar exposure assessment methods; however, some included AD as a secondary outcome and had less statistical power due to smaller sample sizes. Overall, these cross-sectional studies provide mixed evidence regarding the association between air pollution and AD in children; major limitations of these studies include their study design, which prevents causal inference, and self-reported outcomes, which could result in misclassification bias.

Longitudinal studies, which have a stronger methodology for assessing the temporality of exposure and outcomes, conducted with children with AD living in urban settings in South Korea provide evidence for the connection between air pollution and AD symptom severity. Song et al. found a significant association between pruritus severity and ultrafine particulate pollution exposure over 2 months in 41 children with AD [63]. Another study with 21 participants found that a 10 µg/m^3^ increase in the daily mean particulate matter less than 2.5 microns in diameter (PM_2.5_) concentration was associated with 40% increased odds (95% CI: 21–61%) of exacerbation of AD symptoms [71]. These investigations were supported by a longer 18-month study with 22 children with AD [64]. Of note, these studies involved small sample sizes, and misclassification bias may have occurred due to the subjective reporting of AD symptoms.

Infants, who have an immature skin barrier, living in urban environments may be particularly vulnerable to developing AD. Proximity to main roads and increased air pollution exposure during early life was associated with higher prevalence of AD in a German birth cohort study: PM_2.5_ exposure, as determined by land-use regression, was associated with an adjusted relative risk of 1.69 (95% CI: 1.04–2.75) for doctor-diagnosed AD [55]. These results are in alignment with findings from another birth cohort study [56] showing that early life NO_2_ exposure was associated with a higher occurrence of childhood AD: odds ratio = 1.18 (95% CI: 1.00–1.39). Of note, exposures to NO_2_ (odds ratio = 1.35, 95% CI: 1.03–1.78), CO (odds ratio = 1.51, 95% CI: 1.16–1.97), and particulate matter less than 10 microns in diameter (PM_10_) (odds ratio = 1.22, 95% CI: 1.02–1.45) before birth, especially in the first trimester when the fetus is rapidly developing, have been shown to increase the risk for the development of AD before 6 months of age [57,70]. In another study, NO_2_ exposure throughout pregnancy was associated with the onset of childhood AD, and postnatal PM_10_ (odds ratio = 0.82, 95% CI: 0.72–0.94) and NO_2_ (odds ratio = 0.80, 95% CI: 0.65–0.98) exposures were associated with decreased AD remission [74]. Additionally, exposure to combinations of different types of air pollutants at varying concentrations in combination with favorable or non-favorable climatic factors such as temperature and humidity can impact the AD prevalence in children [77].

Regarding indoor air pollution exposure, two studies illustrated positive associations with the AD symptoms. A prospective study conducted with schoolchildren in Seoul, Korea, found that indoor exposure to pollutants, such as toluene, PM_10_, and nitrogen oxide, can increase risk for pruritus in patients with AD [65]. Additionally, a reduction in the indoor PM_10_ concentrations following the implementation of a 7-month school program to improve the indoor air quality was strongly associated with a reduced prevalence of AD and mean eczema area and severity index (EASI) scores [66]. Another study showed that indoor air pollution exposure exacerbated the AD symptoms in children during the spring, winter, and cooler temperatures, with a greater risk in patients with inhalant allergen sensitization and pre-existing severe AD [96].

#### 2.2.2. Air Pollution and Adult AD Prevalence and Incidence

Compared to studies conducted with children, there are fewer studies with adult participants, though they have primarily found positive associations. A survey study in Sweden and case–control study in Taiwan reported that exposure to air pollutants, such as PM_2.5_, was associated with a modestly increased AD prevalence (OR = 1.05, 95% CI: 1.02–1.08) [58,59]. In addition, a study in Australia found positive associations between PM_2.5_ and NO_2_ exposures, as calculated by land-use regression models and self-reported AD symptoms in adult men [76]. A cross-sectional study in the United Kingdom with a similar sample size and exposure assessment technique did not find a significant result for NO_2_, possibly because the outcome assessment was less accurate and did not include the skin prick test results for atopy similar to the Australian study [86].

In a longitudinal study, the incidence of AD symptoms in German women after age 55 was significantly associated with exposure to TRAP, nitrogen oxides, PM_2.5_, and PM_10_, as estimated by monitoring data and land-use regressions [60]. The risk associated with exposure to nitrogen oxides was almost three times higher for minor allele carriers of the aryl hydrocarbon receptor polymorphism rs2066853, which affects the transcriptional activation domain in the *AHR* gene compared to non-carriers, implicating the AhR pathway in adult onset AD associated with TRAP. A follow-up study showed that these associations were stronger in eczema patients without markers for atopy compared to those with atopic markers, suggesting that air pollution may play a bigger role in the development of a nonatopic form of eczema in older individuals, though further research is needed [67].

#### 2.2.3. Air Pollution and Healthcare Utilization for AD

Increasing levels of air pollution can have broader impacts on the healthcare system by driving up service utilization and costs [100]. All 12 studies we found performed an exposure assessment with air pollution concentrations from air monitoring stations and measured clinic, hospital, and emergency department visits for AD based on International Classification of Diseases (ICD-10) codes. Eight epidemiologic studies conducted in China found positive associations between the poor air quality and medical visits for AD among children and adults [78,79,93,94,95,98,101,102], findings that were supported by studies conducted in Korea and Turkey [97,99]. One study used a case-crossover analysis and found that combined adult and pediatric visits for eczema increased by 3.81%, 3.18%, 5.43%, and 5.57% per interquartile range increase in PM_2.5_, PM_10_, NO_2_, and SO_2_ concentrations, respectively [93]. In another study, most air pollutants showed roughly linear exposure–response relationships with total daily AD visits, suggesting the potential for no major threshold effect for air pollution on AD exacerbations [95]. Regarding a period of intense, short-term exposure to air pollution, a recent study found that wildfire air pollution was associated with significantly increased clinic visits for an AD: rate ratio of 1.49 (95% CI: 1.07–2.07) for pediatric patients and 1.15 (95% CI: 1.02–1.30) for adult patients [3]. A follow-up study found that the rates of clinic visits for AD for adults was higher among those age ≥ 65 years of age compared to that of younger adults, which suggests that the skin of older adults has a greater vulnerability to air pollution [80]. In combination, these studies suggest system-level impacts of poor air quality on both pediatric and adult visits for AD at healthcare centers.

## 3. Discussion

The results of in vitro, animal, and epidemiologic studies suggest that exposure to air pollutants increases the risk of the development and exacerbation of AD. Pollution exposure in utero, during childhood, and during adulthood has been shown to contribute to the incidence, prevalence, and severity of AD. The mechanistic pathways underlying the impact of air pollution on skin health include the activation of the AhR pathway, induction of oxidative stress, impairment of the skin barrier, and stimulation of an inflammatory response. Future research could focus on how individuals and mixtures of air pollutants can contribute to AD through the modulation of skin homeostasis and its microbiome alongside other environmental factors, such as ultraviolet radiation. Population-based studies demonstrate mixed results regarding the association between air pollution and AD, but the majority of longitudinal studies, which are more robust than cross-sectional studies, found that exposure to pollutants was positively associated with AD in pediatric and adult populations. Variations in the results from epidemiologic studies can be attributed to several potential factors, including types of air pollutants studied, sources of air pollution, magnitude of exposure, influence of concurrent climatic factors, precision of outcome assessment, and confounders included in statistical analyses. Future epidemiologic studies could be conducted with large, prospective cohorts with both meticulous exposure assessments of multiple air pollutants and precise outcome assessments with physician-reported diagnoses and symptom severity.

The climate crisis has contributed to a recent increase in the occurrence and severity of wildfires in the U.S. and around the world, leading to poor air quality. In California, wildfires burned approximately 1.6 million acres of land in 2018 and 4.3 million acres of land in 2020 [103], and millions of Australians and Brazilians experienced exposure to hazardous levels of air pollutants arising from the wildfires in 2019 and 2020 [104,105]. Since wildfire smoke can travel far distances, it will likely broadly affect patients’ skin health and quality of life [3,106]. This is an important issue for clinicians who counsel patients with AD symptoms during periods of poor air quality as well as public health practitioners who produce public communications and policies that aim to reduce the risks for skin symptom exacerbations. To better understand the effects of air pollution on skin conditions, novel methodologies such as machine learning tree-based models and artificial neural networks are being used to predict AD development with air pollution data [107,108]. Of note, disparities in AD outcomes may be exacerbated by the roles of social determinants of health, such as race, income, education, and geography, in determining exposure to air pollution [109]. Social and structural factors that affect the generation of air pollution, geographic area of residence, and access to healthcare can contribute to environmental health inequities for AD [110,111,112].

As the understanding of air pollution and AD increases, new public health interventions and targeted treatments can be developed. Environmental policies that ban fossil fuel production and use, as well as promote sustainable land management practices, could decrease the amount of outdoor air pollution [113] and could reduce the risk for skin disease exacerbations. In addition, the installation of air filtration devices to improve the air quality in indoor settings could help to reduce the prevalence of AD and severity of the symptoms [66]. Regarding clinical management, specific emollients have been reported to improve the skin barrier function in AD [114,115], which could possibly also reduce pollution-induced symptoms. Additionally, patients could be advised to wear long sleeves and long pants to limit pollution exposure, similar to protection from ultraviolet radiation [116]. However, research on the effectiveness of preventative interventions is needed to develop evidence-based clinical recommendations for topical products and skin protection methods. Of note, topical AhR-modulating medications are being developed and tested for the management of AD and psoriasis, and future research can ascertain whether they can be used to alleviate pollution-related AD symptoms [117]. In addition, recent research has shown how nanomaterials can encapsulate antioxidants and other medications to improve their bioavailability and therapeutic effects [118,119]. Since antioxidants can help reduce oxidative stress and lipid peroxidation induced by environmental exposures, it is possible that this method of drug delivery to the skin may help manage pollution-related skin disease flares [118]. In addition, there are limited clinical studies that have investigated the effectiveness and safety of topical corticosteroids incorporated into lipid nanoparticles for the treatment of eczema [120,121]. Further research is needed to develop treatments that prevent or mitigate air pollution-related skin symptoms, and more robust epidemiological evidence would inform public health education.

## 4. Conclusions

Air pollution is a recognized public health issue, and it has been shown to have several negative impacts on the human body, including skin disease, obstructive pulmonary diseases, cardiovascular diseases, strokes, psychological stress, and poor obstetric outcomes. Overall, the scientific literature evaluated in this review paper found evidence in the support of an association between air pollution exposure and atopic dermatitis. Molecular, cellular, and animal studies demonstrated that the biological underpinnings of the pollution–AD relationship include activation of the AhR pathway, generation of reactive oxidative species, weakening of the skin barrier, and promotion of a proinflammatory response. The epidemiologic evidence was mixed, but most longitudinal studies found that exposure to air pollutants was positively associated with AD in both the adult and pediatric populations. Further areas of research needed in this topic include characterizing air pollution-related inequities in skin disease, air pollutants’ interactions with the skin microbiome, longitudinal exposure to mixtures of air pollutants and the impacts on the incidence and severity of AD, and the effectiveness of clinical interventions in managing air pollution-induced skin exacerbations. 

## Figures and Tables

**Table 1 ijerph-20-02526-t001:** Epidemiology studies showing positive associations between air pollution exposure and atopic dermatitis.

Title	Authors (Year)	Country	Patient Profile	Study Size	Air Pollutants	Exposure Assessment	Outcome Measurement
Environmental Factors, Parental Atopy and Atopic Eczema in Primary-School Children: A Cross-Sectional Study in Taiwan	Lee et al. (2007) [61]	Taiwan	Children (6–12)	10,951 boy and 10,340 girl students whose parents filled out surveys	SO_2_, NO_x_, O_3_, CO, and PM_10_	Air monitoring station data; perceived pollution exposure asked about in surveys	Questionnaire modified from ISAAC asking about patient-reported eczema symptoms
Atopic Diseases, Allergic Sensitization, and Exposure to Traffic-related Air Pollution in Children	Morgenstern et al. (2008) [56]	Germany	Children	2860 children at the age of 4 years and 3061 at the age of 6 years from two prospective birth cohort studies (GINI and LISA)	Traffic-related PM_2.5_ and NO_2_	Exposure modeling (linear models)	Questionnaire asking about doctor-diagnosed eczema and patient-reported eczema symptoms in previous 12 months
Traffic-Related Air Pollution, Climate, and Prevalence of Eczema in Taiwanese School Children	Lee et al. (2008) [62]	Taiwan	Children (mostly 12–14)	158,732 boy and 159,194 girl students whose parents filled out surveys	SO_2_, NO_x_, O_3_, CO, and PM_10_	Air monitoring station data; principal component factor analysis with varimax motion for source-specific exposures	Questionnaire modified from ISAAC asking about patient-reported eczema symptoms
Epidemiology of Eczema among Lebanese Adolescents	Al-Sahab et al. (2008) [68]	Lebanon	Children aged 13–14 years	3153 children	Living near a busy area	ISAAC environmental questionnaire	ISAAC questionnaires
Self-Reported Truck Traffic on the Street of Residence and Symptoms of Asthma and Allergic Disease: A Global Relationship in ISAAC Phase 3	Brunekreff et al. (2009) [54]	Multi-center study around the world (ISAAC Phrase 3)	Children	315,572 children 13–14 years of age from 110 centers in 46 countries and 197,515 children 6–7 years of age from 70 centers in 29 countries	Traffic-related air pollution (TRAP)	Self-reported description of truck traffic on street of residence via surveys	Questionnaires about patient-reported symptoms of eczema within the past 12 months and occurrence of appearing and disappearing rash
Eczema, Respiratory Allergies, and Traffic-Related Air Pollution in Birth Cohorts from Small-Town Areas	Krämer et al. (2009) [55]	Germany	Children	3390 newborns and kids	Traffic-related soot and NO_2_	Land-use regression models	Annual self-reported questionnaires
Acute Health Effects of Urban Fine and Ultrafine Particles on Children with Atopic Dermatitis	Song et al. (2011) [63]	South Korea	Children (8–12)	41 students with AD	PM_10_, PM_2.5_, PM_1_, NO_2_, SO_3_, and O_3_	Air monitoring station data and rooftop spectrometer measurements	Patient diaries, including self-reported eczema severity scores ranging from 0–10
Eczema among Adults: Prevalence, Risk Factors and Relation to Airway Diseases. Results from a Large-Scale Population Survey in Sweden	Rönmark et al. (2012) [59]	Sweden	Adults	18,087 survey respondents	Gas, dust, or fumes exposure at work	Self-reported questionnaire	GA2LEN questionnaire asking about patient-reported eczema symptoms and diagnosis
Symptoms of Atopic Dermatitis Are Influenced by Outdoor Air Pollution	Kim et al. (2013) [64]	South Korea	Children (16–85 months)	17 boys and 5 girls with AD	PM_10_, PM_2.5_, NO, NO_2_, NO_x_, and VOCs	Air monitoring station data	Patient diaries, including eczema severity scores ranging from 0–10
Improvement of Atopic Dermatitis Severity after Reducing Indoor Air Pollutants	Kim et al. (2013) [66]	South Korea	Children (1–5 years)	210 male and 215 female children	PM_10_, CO, CO_2_, and formaldehyde	Air quality monitors	Diagnosis determined by dermatologist examination; surveys: Eczema Area and Severity Index (EASI) and investigator’s global assessment (IGA) measurement
Association between Environmental Factors and Current Asthma, Rhinoconjunctivitis and Eczema Symptoms in School-Aged Children from Oropeza Province—Bolivia: A Cross-Sectional Study	Solis-Soto, Patińo, Nowak, and Radon (2013) [69]	Bolivia	Children aged 9–15 years	2340 children	Intensity of truck traffic near residence	ISAAC environmental questionnaire: frequency of truck traffic	ISAAC questionnaires
Prenatal Air Pollutant Exposure and Occurrence of Atopic Dermatitis	Huang et al. (2015) [57]	21 counties across Taiwan	Children	16,686 mother—infant pairs	NO_2_, SO_2_, CO, O_3_, and PM_10_	Spatial interpolation, GIS, and cross-validation for exposure modeling	Questionnaire filled out by parents about physician-diagnosed AD
Indoor Air Pollution Aggravates Symptoms of Atopic Dermatitis in Children	Kim et al. (2015) [65]	South Korea	Children	30 children with AD	NO, NO_2_, NO_x_, PM_10_, PM_2.5_, PM_1_, and VOCs	Air quality monitors	Teacher recorded pruritus symptoms in diaries for children (0–10)
Association of Pollution and Climate with Atopic Eczema in US Children	Kathuria and Silverberg (2016) [77]	United States	Children	91,642 children	CO, NO_3_, NO_2_, OC, SO_3_, SO_2_, PM_2.5_, PM_10_, and O_3_	Monitoring systems	National Survey of Children’s Health questionnaire
Adult Atopic Dermatitis and Exposure to Air Pollutants—A Nationwide Population-Based Study	Tang et al. (2017) [58]	Taiwan	Adults	1023 patients with AD and 4092 controls	PM_2.5_ and the Pollutant Standards Index (PSI)	Data from ground-level monitoring stations	Physician-diagnosed AD
Association between Exposure to Traffic-Related Air Pollution and Prevalence of Allergic Diseases in Children, Seoul, Korea	Yi et al. (2017) [75]	South Korea	Children	14,756 children	Traffic-related air pollution (TRAP)	Road network data on proximity to and density of major roads	Questionnaire modified from ISAAC
Traffic-Related Air Pollution and Eczema in the Elderly: Findings from the SALIA Cohort	Schnass et al. (2018) [60]	West Germany	Adult women aged 55+	834 women from the SALIA cohort	Traffic-related air pollution (NO_2_ and NO_x_), PM_2.5_, PM_coarse_, and PM_10_	Monitoring data, back-extrapolation algorithm, and land-use regressions	Questionnaire modified from ISAAC asking about patient-reported eczema symptoms and physician-diagnosed eczema
Preventive Effect of Residential Green Space on Infantile Atopic Dermatitis Associated with Prenatal Air Pollution Exposure	Lee et al. (2018) [70]	South Korea	Pregnant women and their babies at age 6 months	659 mothers and their babies	Exposure to traffic-related air pollution: PM_10_, and NO_2_	Land use regression models with data from air monitoring stations	ISAAC questionnaires
Association between Particulate Matter Concentration and Symptoms of Atopic Dermatitis in Children Living in an Industrial Urban Area of South Korea	Oh et al. (2018) [71]	South Korea	Children aged 1–5 years	21 children with AD	PM_10_ and PM_2.5_	Air quality monitoring stations	Physician-confirmed diagnosis and parent-recorded symptom diary
Nonatopic Eczema in Elderly Women: Effect of Air Pollution and Genes	Hüls et al. (2019) [67]	Germany	Adult women aged 55+	834 women from the SALIA cohort	NO_2_, NO_x_, PM_2.5_, and PM_10_	Monitoring data, back-extrapolation algorithm, and land-use regressions	Questionnaire modified from ISAAC asking about patient-reported eczema symptoms and physician-diagnosed eczema
Ambient Air Pollution and the Hospital Outpatient Visits for Eczema and Dermatitis in Beijing: A Timestratified Case-Crossover Analysis	Guo et al. (2019) [93]	China	Children and adults	157,595 visits	PM_2.5_, PM_10_, NO_2_, and SO_2_	Air quality monitoring stations	Clinic and hospital visits based on International Classification of Diseases (ICD) codes
Association between Exposure to Traffic-Related Air Pollution and Pediatric Allergic Diseases Based on Modeled Air Pollution Concentrations and Traffic Measures in Seoul, Korea: A Comparative Analysis	Min et al. (2020) [73]	South Korea	Children aged 1–12 years	14,614 children	PM_2.5_, PM_10_, and NO_2_	Air quality monitoring sites and prediction models for NO_2_, PM_10_ and land use regressions for PM_2.5_	ISAAC questionnaires
Relative Impact of Meteorological Factors and Air Pollutants on Childhood Allergic Diseases in Shanghai, China	Hu et al. (2020) [94]	China	Children	787,646 cases	PM_2.5_, PM_10_, NO_2_, O_3_, and SO_2_	Air quality monitoring stations	Clinic and hospital visits based on ICD codes
Association between Air Pollution and Atopic Dermatitis in Guangzhou, China: Modification by Age and Season	Wang et al. (2020) [95]	China	Children and adults	29,972 visits	PM_2.5_, PM_10_, NO_2_, O_3_, and SO_2_	Air quality monitoring stations	Clinic visits based on ICD codes
Association between Ambient Air Pollution and Development and Persistence of Atopic and Non-Atopic Eczema in a Cohort of Adults	Lopez et al. (2021) [76]	Australia	Adults	2369 adults	PM_2.5_ and NO_2_	Satellite-based land-use regression model	Self-administered postal survey (questionnaire and skin prick test results)
Exposure to Air Pollution and Incidence of Atopic Dermatitis in the General Population: A National Population-Based Retrospective Cohort Study	Park, Kim, and Seo (2021) [72]	South Korea	Children and adults	209,168 people without AD at start of study; 3203 developed AD	PM_10_, PM_2.5_, SO_2_, NO_2_, O_3_, and CO	Air quality monitoring stations	ICD-10 code from insurance database
Effects of Exposure to Indoor Fine Particulate Matter on Atopic Dermatitis in Children	Kim et al. (2021) [96]	South Korea	Children	64 children	PM_2.5_	Indoor laser-based air quality sensor	Physician-confirmed diagnosis and Atopic Dermatitis Symptom Score (ADSS)
Onset and Remission of Eczema at Pre-School Age in Relation to Prenatal and Postnatal Air Pollution and Home Environment across China	Lu et al. (2021) [74]	China	Children	39,782 children	PM_2.5_, PM_10_, and NO_2_	Monitoring station data and inverse distance weighted air pollution models	ISAAC questionnaires
Effects of Climate and Air Pollution Factors on Outpatient Visits for Eczema: A Time Series Analysis	Karagün, Yildiz, and Cangür (2021) [97]	Turkey	Children and adults	27,549 patients	PM_10_ and SO_2_	Air quality monitoring stations	Clinic visits based on ICD codes
NO_2_ Exposure Increases Eczema Outpatient Visits in Guangzhou, China: An Indication for Hospital Management	Zhang et al. (2021) [98]	China	Children and adults	293,000 patients	PM_2.5_, PM_10_, NO_2_, O_3_, and SO_2_	Air quality monitoring stations	Clinic visits based on ICD codes
Associations between Ambient Air Pollution and Medical Care Visits for Atopic Dermatitis	Baek, Cho, and Roh (2021) [99]	South Korea	Children and adults	513,870 visits	PM_2.5_, PM_10_, NO_2_, O_3_, CO, and SO_2_	Air quality monitoring stations	Clinic, hospital, and emergency department visits based on ICD codes
Association of Wildfire Air Pollution and Health Care Use for Atopic Dermatitis and Itch	Fadadu et al. (2021) [3]	United States	Children and adults	8049 visits; 4174 patients	PM_2.5_ and wildfire smoke	Air quality monitoring stations and satellite imagery	Clinic visits based on ICD codes
Air Pollution and Weather Conditions Are Associated with Daily Outpatient Visits of Atopic Dermatitis in Shanghai, China	Ye at al. (2022) [78]	China	Children and adults	34,633 patients	PM_2.5_, PM_10_, NO_2_, O_3_, and SO_2_	Air quality monitoring stations	Clinic visits based on ICD codes
Relationship between Air Pollution and Childhood Atopic Dermatitis in Chongqing, China: A Time-Series Analysis	Luo et al. (2022) [79]	China	Children	214,747 patients	PM_2.5_, PM_10_, SO_2_, NO_2_, O_3_, and CO	Air quality monitoring stations	Clinic visits based on ICD codes
Association of Exposure to Wildfire Air Pollution With Exacerbations of Atopic Dermatitis and Itch Among Older Adults	Fadadu et al. (2022) [80]	United States	Children and adults	5529 visits; 3448 patients	PM_2.5_ and wildfire smoke	Air quality monitoring stations and satellite imagery	Clinic visits based on ICD codes

Abbreviations: AD, atopic dermatitis; PM_2.5_, particulate matter less than 2.5 microns in diameter; PM_10_, particulate matter less than 10 microns in diameter; PM_1_, particulate matter less than 1 micron in diameter; O_3_, ozone; CO, carbon monoxide; OC: organic carbon; SO_2_, sulfur dioxide; NO_3_: nitrate; NO_2_, nitrogen dioxide; NO, nitric oxide; ISAAC, International Study of Asthma and Allergies in Childhood; VOC, volatile organic compounds; ICD: International Classification of Diseases.

**Table 2 ijerph-20-02526-t002:** Epidemiology studies showing null or inconclusive associations between air pollution exposure and atopic dermatitis.

Title	Authors (Year)	Country	Patient Profile	Study Size	Air Pollutants	Exposure Assessment	Outcome Measurement
Long-term Exposure to Background Air Pollution Related to Respiratory and Allergic Health in Schoolchildren	Pénard-Morand et al. (2005) [85]	France	Children	6620 children from 108 schools	NO_2_, SO_2_, PM_10_, and O_3_	3-year-averaged concentrations of air pollutants using background monitoring stations; Low or High classification	Skin examination for flexural dermatitis and AD assessed with standardized health questionnaire completed by parents (ISAAC)
Traffic-Related Air Pollution and the Development of Asthma and Allergies during the First 8 Years of Life	Gehrig et al. (2009) [83]	Netherlands	Children	3863 children in the PIAMA birth cohort study	NO_2_, PM_2.5_, and soot	Land-use regression models	Parental-reported questionnaires on doctor-diagnosed AD
Effect of Traffic Pollution on Respiratory and Allergic Disease in Adults: Cross-Sectional and Longitudinal Analyses	Pujades-Rodríguez et al. (2009) [86]	United Kingdom	Adults (18–70)	2599 adults	NO_2_	Grid-based exposure modelling	Questionnaire asking about physician-diagnosed AD
Ambient Particulate Pollution and the World-Wide Prevalence of Asthma, Rhinoconjunctivitis and Eczema in Children: Phase One of the International Study of Asthma and Allergies in Childhood (ISAAC)	Anderson et al. (2009) [89]	Multi-center study around the world (ISAAC Phase 1)	Children aged 6–7 years and 13–14 years	190,624 children aged 6–7 years and 322,529 children aged 13–14 years	PM_10_	City level annual concentrations based on the World Bank model	ISAAC questionnaires
Which Population Level Environmental Factors Are Associated with Asthma, Rhinoconjunctivitis and Eczema? Review of the Ecological Analyses of ISAAC Phase One	Asher et al. (2010) [82]	Multi-center study around the world (ISAAC Phase 1)	Children aged 6–7 years and 13–14 years	463,801 children aged 13–14 years across 56 countries, and in 257,800 children aged 6–7 years across 38 countries	PM_10_	Used the World Bank Global Model on Ambient Particulates for 1999 to estimate annual concentrations	Surveys asking about eczema symptoms within the last 12 months
Early-life Exposure to Outdoor Air Pollution and Respiratory Health, Ear Infections, and Eczema in Infants From the INMA Study	Aguilera et al. (2013) [81]	Spain	Infants	2199 infants in a population-based birth cohort	NO_2_ and benzene	Land use regression models	Parent-reported via questionnaires (did not ask if symptoms were specifically doctor-diagnosed)
Allergens, Air Pollutants, and Childhood Allergic Diseases	Wang, Tung, Tang, and Zhao (2016) [88]	Taiwan	Kindergarten children	2661 children	PM_10_, PM_2.5_, NO_2_, and O_3_	Data from monitoring stations	ISAAC questionnaires
The Effects of PM_2.5_ on Asthmatic and Allergic Diseases or Symptoms in Preschool Children of Six Chinese Cities, Based on China, Children, Homes and Health (CCHH) Project	Chen et al. (2018) [87]	China	Children (mean age of 4.6 years)	30,759 children	PM_2.5_ and O_3_	Using an exposure database that combines satellite data, transport models, and ground measurements	Core questionnaire of ISAAC
Atopic Dermatitis: Interaction Between Genetic Variants of GSTP1, TNF, TLR2, and TLR4 and Air Pollution in Early Life	Hüls et al. (2018) [84]	Sites in Canada and Europe	Children	6 birth cohorts: 5685 participants (TAG study)	NO_2_ and traffic-related air pollution	Land-use regression models and dispersion modeling	Parental-reported questionnaires on doctor-diagnosed AD and AD symptoms
Eczema, Facial Erythema, and Seborrheic Dermatitis Symptoms among Young Adults in China in Relation to Ambient Air Pollution, Climate, and Home Environment	Wang et al. (2021) [90]	China	Kindergarten children	40,279 respondents to surveys	PM_10_ and NO_2_	City level annual concentration from air monitoring stations	Surveys in the China, Children, Homes, and Health study

Abbreviations: AD, atopic dermatitis; PM_2.5_, particulate matter less than 2.5 microns in diameter; PM_10_, particulate matter less than 10 microns in diameter; O_3_, ozone; SO_2_, sulfur dioxide; NO_2_, nitrogen dioxide; ISAAC, International Study of Asthma and Allergies in Childhood.

## Data Availability

Not applicable.

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
