# Peer review of "Air Pollution and Atopic Dermatitis, from Molecular Mechanisms to Population-Level Evidence: A Review"

_ijerph, 2023, doi:10.3390/ijerph20032526_

Round 1
Reviewer 1 Report
The paper investigates, as a wide literature's review, the role of pollution on atopic dermatitis.
The authors describes the underlying pathogenetic mechanisms about the development and the worsening of AD.
Conclusions are interesting when suggest advice for clinicians and when describe worsening of disease during major pollution periods.
Tables are clear and well done.
Author Response
Many thanks to Reviewer 1 for the positive comments.
Reviewer 2 Report
The review article entitled "Air pollution and atopic dermatitis, from molecular mechanisms to population-level evidence" highlights the most neglected environmental risk to public health, particularly in industrialized regions of the world. The article is comprehensively written and provides deep insights into skin infections getting worse due to decreasing air quality.
The approach of the authors is valid and the article is well-written providing scientific evidence from published research. I think the article can be accepted after a little improvement in the conclusion part. The conclusion may include some other effects on public health related to air pollution.
Author Response
Many thanks to the reviewers for taking the time to provide their thoughtful comments. We have carefully addressed each recommendation in itemized responses below, with italicized text modified in the manuscript and line numbers referring to the marked version. We think that the manuscript is strengthened by these changes.
Reviewer #2:
The review article entitled "Air pollution and atopic dermatitis, from molecular mechanisms to population-level evidence" highlights the most neglected environmental risk to public health, particularly in industrialized regions of the world. The article is comprehensively written and provides deep insights into skin infections getting worse due to decreasing air quality. The approach of the authors is valid and the article is well-written providing scientific evidence from published research.
- I think the article can be accepted after a little improvement in the conclusion part. The conclusion may include some other effects on public health related to air pollution.
Thank you for this suggestion. We have added the health effects of wildfire smoke on other organ systems in the new conclusion section, lines 396-399: “Air pollution is a recognized public health issue, and wildfire smoke has been shown to have several negative impacts on the human body, including skin disease, obstructive pulmonary diseases, cardiovascular diseases, strokes, psychological stress, and poor obstetric outcomes.”
Reviewer 3 Report
1.Please add previous in vitro/in vivo studies about the toxicities.
2.It will be great if authors add section about the in vitro effect of novel nanomaterials in air pollution induced Atopic dermatitis.
3.Please add your future perspective and suggestion about the possible role of antioxidant therapy against pollution related atopic dermatitis
4.It is suggested to use these papers for update discussion part and bold the novelty of your study :
-DOI: 10.1016/j.biopha.2018.04.126
-DOI: 10.1155/2021/4946711
-DOI: 10.1016/j.pestbp.2020.104586
Author Response
Many thanks to the reviewers for taking the time to provide their thoughtful comments. We have carefully addressed each recommendation in itemized responses below, with italicized text modified in the manuscript and line numbers referring to the marked version. We think that the manuscript is strengthened by these changes.
Reviewer #3:
1.Please add previous in vitro/in vivo studies about the toxicities.
We have added an additional toxicology study regarding the toxicology mechanism of PAHs in the generation reactive oxygen species, lines 134-136: “One mechanism involves intracellular metabolic pathways that convert PAHs into quinones, generating superoxide anion, hydrogen peroxide, and ROS.32” and incorporated this into the Molecular Pathogenesis section, where we have reviewed and summarized the main findings of key in vitro and in vivo studies performed on air pollution and atopic dermatitis.
2.It will be great if authors add section about the in vitro effect of novel nanomaterials in air pollution induced Atopic dermatitis.
Thank you for the suggestion. We have included information regarding how nanoparticles promote inflammation in the skin based on in vitro studies in the 2.1.4 Inflammation section, lines 186-189: “In addition, exposure of skin cells to airborne nanoparticles, such as carbonaceous pollutants, diesel exhaust particles, and tungsten carbide cobalt particles, can induce production of pro-inflammatory cytokines and alter several cellular signaling pathways.52,53”
3.Please add your future perspective and suggestion about the possible role of antioxidant therapy against pollution related atopic dermatitis
We have added additional information about antioxidant therapy for the treatment of air pollution-related atopic dermatitis as well as drug delivery through the use of nano-particles in the discussion, lines 384-390: “In addition, recent research has shown how nano-materials can encapsulate antioxidants and other medications to improve their bioavailability and therapeutic effects.118,119 Since antioxidants can help reduce oxidative stress and lipid peroxidation induced by environmental exposures, it is possible that this method of drug delivery to the skin may help manage pollution-related skin disease flares.118 In addition, there are limited clinical studies that have investigated the effectiveness and safety of topical corticosteroids incorporated into lipid nanoparticles for the treatment of eczema.120,121 Further research is needed to develop treatments that prevent or mitigate air pollution-related skin symptoms…”
4.It is suggested to use these papers for update discussion part and bold the novelty of your study :
-DOI: 10.1016/j.biopha.2018.04.126
-DOI: 10.1155/2021/4946711
-DOI: 10.1016/j.pestbp.2020.104586
Thank you for the suggested references. We have incorporated 2 of these references, lines 384-386: “In addition, recent research has shown how nano-materials can encapsulate antioxidants and other medications to improve their bioavailability and therapeutic effects.118,119”
Reviewer 4 Report
The manuscript entitled “Air pollution and atopic dermatitis, from molecular mechanisms to population-level evidence: a narrative review” is interesting. Some suggestions are as followed to further improve the content:
1. Line 31- The definition of air pollution is misleading. Kindly refine the sentence, especially mixture of gaseous molecules. Not all gaseous molecules are pollutant.
2. Some details on airborne solid particles/ pollutants shall be discussed since authors mentioned this in line 31. Attached some recent articles that related to airborne particles for the authors reference and authors could retrieve some info and added into the introduction section.
https://doi.org/10.1016/j.buildenv.2022.109489
https://doi.org/10.1080/17512549.2021.2009911
https://doi.org/10.1007/s10973-022-11466-6
https://doi.org/10.1016/j.enbuild.2022.112277
3. Line 56-line 61- Would suggest the authors to systematically highlight the goal/ objectives. For example, the objectives of this manuscript are: (i) xxxxx, (ii) xxxxx, (iii) xxx.
4. In table 1, under air pollutant row, I suggest authors to write the pollutant in full, instead of TRAP.
5. Line 214- A comprehensive statistics for infants living in urban environment that developed AD shall be included.
6. Line 254- I don’t suggest authors just write “rs2066853”. Make it into layman term such as hydrocarbon receptor XXX shall be included to assist the readers.
7. Suggest splitting discussion and conclusion sections. The conclusion should be comprehensive, which address the objectives.
Author Response
Many thanks to the reviewers for taking the time to provide their thoughtful comments. We have carefully addressed each recommendation in itemized responses below, with italicized text modified in the manuscript and line numbers referring to the marked version. We think that the manuscript is strengthened by these changes.
Reviewer #4:
The manuscript entitled “Air pollution and atopic dermatitis, from molecular mechanisms to population-level evidence: a narrative review” is interesting. Some suggestions are as followed to further improve the content:
- Line 31- The definition of air pollution is misleading. Kindly refine the sentence, especially mixture of gaseous molecules. Not all gaseous molecules are pollutant.
Thank you for this suggestion. We have modified the sentence as follows, lines 31-32: “Air pollution is a complex mixture that includes certain solid particles, liquid droplets, and gaseous molecules.”
- Some details on airborne solid particles/ pollutants shall be discussed since authors mentioned this in line 31. Attached some recent articles that related to airborne particles for the authors reference and authors could retrieve some info and added into the introduction section.
https://doi.org/10.1016/j.buildenv.2022.109489
https://doi.org/10.1080/17512549.2021.2009911
https://doi.org/10.1007/s10973-022-11466-6
https://doi.org/10.1016/j.enbuild.2022.112277
Thank you for the suggestion to expand the details regarding airborne solid particles/pollutants. We have added information from the US Environmental Protection Agency website on air pollution, lines 35-36: “Solid particles, often including ultra-fine particles smaller than 0.1 microns, are found in dust, smoke, and soot.1 ”
Thanks also for directing us to the interesting papers on mitigation of airborne infectious particles in the operating room setting. Unfortunately, the inclusion of infectious particles is beyond the scope of our review, which is focused on nonbiological air pollution generally produced by industrial, traffic, and wildfire sources.
3. Line 56-line 61- Would suggest the authors to systematically highlight the goal/ objectives. For example, the objectives of this manuscript are: (i) xxxxx, (ii) xxxxx, (iii) xxx.
Thank you for this good idea on how to improve the organization of the paper. We have updated the last sentence of the Introduction to clearly define the goals of the review article, lines 78-82: “The aims of this manuscript are to discuss the connection between air pollution and AD through (i). reviewing potential biological mechanisms, (ii). evaluating population-level evidence, including impacts on healthcare utilization that were not included in previous reviews, and (iii). discussing potential avenues for future research.”
4. In table 1, under air pollutant row, I suggest authors to write the pollutant in full, instead of TRAP.
In Table 1 and Table 2, we have removed the acronym TRAP from the air pollution column, and instead have written out “traffic-related air pollution.”
5. Line 214- A comprehensive statistics for infants living in urban environment that developed AD shall be included.
Unfortunately, we were unable to locate a comprehensive statistic after a literature search. If the Reviewer is aware of a publication with this statistic, we welcome the reference and will include it in the paper. We direct the Reviewer to the lines 265-268: “Proximity to main roads and increased air pollution exposure during early life was associated with higher prevalence of AD in a German birth cohort study: PM2.5 exposure, as determined by land-use regression, was associated with an adjusted relative risk of 1.69 (95% CI: 1.04-2.75) for doctor-diagnosed AD.”
- Line 254- I don’t suggest authors just write “rs2066853”. Make it into layman term such as hydrocarbon receptor XXX shall be included to assist the readers.
We have clarified that rs2066853 is a polymorphism of the AHR gene, lines 303-304: “The risk associated with exposure to nitrogen oxides was almost three times higher for minor allele carriers of the aryl hydrocarbon receptor polymorphism rs2066853, which affects the transcriptional activation domain in the AHR gene, than non-carriers…”
- Suggest splitting discussion and conclusion sections. The conclusion should be comprehensive, which address the objectives.
Thank you for this suggestion. We have created a new section called “Conclusion” that summarizes the main points related to the objectives of the review, with the first sentence added to address Reviewer2’s item 1, lines 396-411: “Air pollution is a recognized public health issue, and wildfire smoke has been shown to have several negative impacts on the human body, including skin disease, obstructive pulmonary diseases, cardiovascular diseases, strokes, psychological stress, and poor obstetric outcomes. Overall, the scientific literature evaluated in this review paper found evidence in support of an association between air pollution exposure and atopic dermatitis. Molecular, cellular, and animal studies demonstrated that the biological underpinnings of the pollution-AD relationship include activation of the AhR pathway, generation of reactive oxidative species, weakening of the skin barrier, and promotion of an inflammatory response. Epidemiologic evidence was mixed, but most longitudinal studies found that exposure to air pollutants was positively associated with AD in both adult and pediatric populations. Further areas of research needed in this topic include characterizing air pollution-related inequities in skin disease, air pollutants’ interactions with the skin microbiome, longitudinal exposure to mixtures of air pollutants and the impacts on incidence and severity of AD, and the effectiveness of clinical interventions to manage pollution-induced skin exacerbations.”
Round 2
Reviewer 4 Report
The authors have addressed most of the questions relatively well. However, I believe that the response statement for Q5 shall be revised. Would suggest the authors performed a touch-up on the paragraph that related to this statement.
Author Response
Many thanks for the further reviewer comments. We have carefully addressed the recommendation below, with italicized text modified in the manuscript and line numbers referring to the marked version.
Reviewer 4:
The authors have addressed most of the questions relatively well. However, I believe that the response statement for Q5 shall be revised. Would suggest the authors performed a touch-up on the paragraph that related to this statement.
Thanks to the reviewer for the opportunity to strengthen and add clarity to the paragraph about AD risk for infants living in urban environments. We have added specific information about the measure of association reported in the cited studies to strengthen the paragraph, and to increase our reporting specificity, lines 265-278: “Proximity to main roads and increased air pollution exposure during early life was associated with higher prevalence of AD in a German birth cohort study: PM2.5 exposure, as determined by land-use regression, was associated with an adjusted relative risk of 1.69 (95% CI: 1.04-2.75) for doctor-diagnosed AD.55 These results are in alignment with findings from another birth cohort study56 showing that early life NO2 exposure was associated with higher occurrence of childhood AD: odds ratio = 1.18 (95% CI: 1.00-1.39). Of note, exposures to NO2 (odds ratio = 1.35, 95% CI: 1.03-1.78) , CO (odds ratio = 1.51, 95% CI: 1.16-1.97), and particulate matter less than 10 microns in diameter (PM10) (odds ratio = 1.22, 95% CI: 1.02-1.45) before birth, especially in the first trimester when the fetus is rapidly developing, have been shown to increase risk for the development of AD before 6 months of age.57,70 In another study, NO2 exposure throughout pregnancy was associated with onset of childhood AD, and postnatal PM10 (odds ratio = 0.82, 95% CI: 0.72-0.94) and NO2 (odds ratio = 0.80, 95% CI: 0.65-0.98) exposures were associated with decreased remission.74”